# Power Output and Downstream Wake Modifications of Floating Turbines Subjected to Ocean Motions Using a Tension-Leg Platform

Juan M. Restrepo<sup>1, 2</sup>, Matthew Norman<sup>1</sup>, Stuart Slattery<sup>1</sup>, Lawrence Cheung<sup>3</sup>, and Yihan Liu<sup>4</sup>

**Correspondence:** Juan M. Restrepo (restrepojm@ornl.gov)

Abstract. A concern in the deployment of large wind turbines on ocean floating platforms is the effect of floating platform motions on their electrical power generation. Further, it is not clear how floating motions influence waking, which might affect the combined power generation of collections of turbines. We examine the average power output of a single and a collection of 5 MW wind turbines, mounted on a Tension-Leg Platform (TLP) under the action of fully developed ocean wave motions, coupling floating motions with Large Eddy Simulation (LES) of atmospheric and rotor dynamics. The ocean dynamics enter as fully developed waves derived from the Pierson-Moskowitz spectrum. To assess the influence of ocean motions we performed simulations under multiple turbulence intensities, reporting comparisons of average power output when the platforms are allowed to move to when they are held rigidly in place. In all simulations, we find that the effects of TLP floating platform's induced motions have a minor effect on single and multiple turbine power production and wake structure. Even when using coherent and large amplitude harmonic floating induced perturbations, any significant wake modifications from floating motions are confined to the near-wake region, where downstream turbines are unlikely to be located. The relatively small amplitude of TLP motions relative to pre-existing turbulent fluctuations are the primary reason for low wake and power modifications downstream.

Copyright statement. This manuscript has been authored by UT-Battelle, LLC under Contract No. DE- AC05-00OR22725 with the U.S. Department of Energy. The publisher, by accepting the article for publication, acknowledges that the U.S. Government retains a non-exclusive, paid up, irrevocable, world-wide license to publish or reproduce the published form of the manuscript, or allow others to do so, for U.S. Government purposes. The DOE will provide public access to these results in accordance with the DOE Public Access Plan (http://energy.gov/downloads/doe-public-access-plan).

#### 1 Introduction

Modeling and simulation of wind turbines has proven to be an effective way to inform the design, siting, and in the prediction of power production (e.g. Wu and Porté-Agel (2015), Miller et al. (2013)). These calculations model individual turbines, as well

<sup>&</sup>lt;sup>1</sup>Oak Ridge National Laboratory, Oak Ridge TN USA 37831

<sup>&</sup>lt;sup>2</sup>Department of Mathematics, University of Tennessee Knoxville, Knoxville TN USA 37920

<sup>&</sup>lt;sup>3</sup>Sandia National Laboratory, Livermore, CA USA 94550

<sup>&</sup>lt;sup>4</sup>Virginia Tech, Blacksburg, VA USA 24061



as the interaction of turbine wakes and its environment. Far offshore settings for power farms are of interest due to the more significant wind resource. Many of these locations are in deep water and may require the turbines to be mounted on floating platforms. This in turn leads us to ask whether ocean states will affect power production of single and multiple turbines and whether wave motions affect the power output positively or otherwise.

An ocean platform would be exposed to a variety of different waves. Wind waves that equilibrate spectrally broadly define 'fully developed' ocean conditions (see Pierson and Moskowitz (1964)). Equilibration depends on fetch conditions and the length of time that steady wind is stressing the sea surface. A fully developed ocean wind wave spectrum has an effective random phase field, and empirical models, such as the Pierson-Moskowitz model and its various specializations (e.g. JONSWAP), are recognized as good models of the fully developed ocean wind wave conditions. Swell waves, generated by distant storms and disturbances, on the other hand, produce episodic/transient changes in the sea surface. Finally, there are white-capping waves, typified by random breaking events, and steep episodic breaking waves of large amplitude. Our focus is on capturing ocean effects due to fully developed ocean conditions, which on daily time scales, are most relevant to estimating power output in floating platform ocean turbines.

Modeling of wind turbines is a multi-scale, multi-physics problem (see Lemmer et al. (2018) and Wang et al. (2022) and references contained therein). The work of Wang et al. (2022) focuses on simulations that form the basis for the Offshore Code Comparison Projects, which use NREL's OpenFAST. Blade-resolved simulations must include boundary layer physics at very small scales while regional characterization of the wind resource requires encapsulation of hour and kilometer-scale phenomena in mesoscale simulations (i.e. simulations that bridge the scales between highly resolved models and earth system simulations). The complexity and multitude of design considerations associated with the various platform configurations has shown the need for comprehensive studies on the modeling offshore wind platforms and their interactions with the environment. The Offshore Code Comparison Collaboration, Continued with Correlation and unCertainty (OC6) project addresses this need for model validation and comparison. Of particular interest to the current work are Phase III and Phase IV (Bergua et al., 2023; Cioni et al.) of the OC6 project, which studied the aerodynamic rotor behavior and full-system, aerodynamic and hydrodynamic interactions of floating platforms. Their results have shown that two-phase flow simulations of semi-submersible and TetraSpar floating platforms (Darling et al., 2025) are able to be successfully compared against experiments, but there are many remaining challenges in these high-fidelity simulations and simplified models of their behavior are still necessary in many instances. Our long-term goal is to create more accurate representations of wind farms in the mesoscale simulations. For this purpose we argue that large eddy simulations (LES) are appropriate to resolve atmospheric interactions with turbines, and the momentum and energetic exchanges critical to power generation by the turbines and their effect on the atmospheric boundary layer. LES has been used to compare the effect of moving platform turbine wakes to those of a fixed platform. For example Johlas et al. (2019, 2020) found the wakes of these two configurations not very dissimilar.

Large-scale simulations using LES of wind farms are now possible with modern computing tools Cheung et al. (2022). These studies allow for computation at different atmospheric conditions and often are able to connect to experimental data Kumer et al. (2016). Unlike blade-resolved models where individual turbines are largely represented in their geometric entirety (see Ribeiro et al. (2023)), LES models often represent the turbine via either an actuator line or actuator disk model Martinez et al. (2012).





The choice of the wind turbine rotor model in high fidelity wind farm simulations is dependent on both the computational resources available and the expected wake dynamics to be encountered. For instance, in floating turbines with a surge motion frequency commiserate with the rotor speed, a vortex ring state can be encountered if the blade tips interact with the vortex emerging from the previous blade (Sebastian and Lackner, 2011; Kyle et al., 2020), and consequently change the near field wake. In other cases, an asymmetric loaded rotor (Abraham and Leweke, 2023) or individual pitch control (Cheung et al., 2024) can lead to different loading on individual blades and change the downstream wake behavior. For those situations, an actuator line method or a blade-resolved method may be appropriate to resolve the tip vortex behavior or blade loading differences. However, differences in wake behavior can also be caused by changes to the turbine orientations or to rotor behavior which are not depending on resolving the individual blade loading profiles or tip vortices. For instance, wake deflections due to turbine yaw steering (Frederik et al., 2025; Brown et al., 2025) or platform yaw or pitch are a steady phenomenon which can directly impact wind farm performance. Previous studies by Li et al. (2022) and Messmer et al. (2025) have also shown that lower-frequency floating platform motions can introduce large scale structures in the far wake which can enhance wake recovery. Such effects can be captured using simpler actuator disk models without the need for turbine or blade pitch controllers and are the focus of the current study.

In the context of LES for mesoscale models, approaches include using LES data to build models for hub height wind speeds and other relevant model parameters Pan and Archer (2018) for static turbine configurations. It is therefore unclear from the literature what effect, if any, moving turbines will have on key performance parameters in mesoscale simulations, particularly at the timescales of hours and days over which mesoscale simulations operate. Inclusion of platform motion in these models has become more prevalent in the literature recently (e.g. Li et al. (2024); Wena et al. (2018); Yang et al. (2022)) but the coupled dynamics of platforms, atmospheric flows and turbines are new.

A significant effort is being made to model and simulate turbine dynamics on moving platforms. These model and simulation efforts are aimed toward discerning first order dynamic/structural issues that will inform the design and construction of these FOWTs. For example, Johlas et al. (2020) compare wake characteristics of turbines mounted on different floating platforms, and their waks to those of fixed turbines.

To model floating platform effects initially at an intermediate resolution using 10m grid spacing to better understand the dynamics relevant to mesoscale, our approach is to first extend LES simulations to include floating turbine dynamics such that power production and subsequently turbulent kinetic energy (i.e. the quantities of interest in a mesoscale parameterization), are directly modified by the turbine motions. To do this, we have modified an existing LES simulation tool capable of modeling a variety of atmospheric conditions using an actuator disk approach. We assess a variety of ocean conditions in the form of wave spectra using both quasi standard distribution induced motions as well as spectra representing harmonic induced motions used more recently in the literature (see Li et al. (2024)). This approach allows us to understand how floating platform motions from ocean dynamics and turbulent thrust intermittency alters key elements of the flow used in mesoscale parameterizations including hub-height wind speeds and directions, turbulence, and turbine power output over the time scales present in a mesoscale simulation. Our approach distinguishes itself from the existing literature in its breadth, by the dynamic coupling of turbine/at-


mosphere and turbines, consideration of different atmospheric turbulent conditions, and consideration of single turbines as well as simple cluster arrangements of multiple turbines.

Recently there has been a dearth of papers that concern the coupled dynamics of floating turbines and the atmospheric boundary layer that they occupy or are planned to occupy.

Floating motions of platforms affect the dynamics of the turbines attached to them. The wind tunnel experiments inWei and Dabiri (2022) report decreased mean power at high tip-speed ratios and high surge due to the stall onset on blades. They were also able to obtain an increase in power under certain controlled wave conditions. This work points to the importance of having active and passive wave motion control on these FOWTs.

Driven primarily by the oil industry, the technological development of ocean platforms has generated very sophisticated designs. Ocean platforms can be actively driven, they might be buoyantly floating moored designs, and most designs have some sort of active ocean motion control. We focus on a specific and familiar moored platform, namely the Tension-Leg Platform, which is well suited to carry a 5MW turbine. However, we deliberately omit active motion control, in the hopes of assessing, as a first pass, how a simple chain anchor system handles ocean dynamics. Further, active platform stability mechanisms are focused on addressing transient motions due to storms, swell, and developing ocean conditions, the residual flow due to waves (see McWilliams and Restrepo (1999); McWilliams et al. (2004); Restrepo (2007)) and thus the results of this study do not extend to these ocean conditions.

The remainder of the document is organized as follows. In Section 2 we detail the model of the platform/turbine dynamics, responding to wind wave motions. In Section 3 we give the problem statement and in Section 4 we present our LES formulation. In Section 5 we describe the results of our numerical experiments and in Section 6 we give our concluding remarks.

# 110 2 Platform Dynamics

Our modeling approach for the simulation of the floating offshore wind turbines dynamics is adapted from Betti et al. (2014). A schematic of the platform and turbine appear in Figure 1. In what follows we will be specifically using the parameters associated with a 5 MW turbine, mounted on a floating platform. Generally we would have 12 degrees of freedom (DOFs): surge, heave, and pitch position, which are the symmetric motions; and the antisymmetric motions are: sway, roll, yaw position (the other six DOFs are the velocities of each aforementioned position). In what follows, however, we will always line up the platforms/turbines such that only symmetric degrees need to be considered (see Ribeiro et al. (2023) and Fontanella et al. (2025) for studies that focus on the effect of sway suge and yaw effects on turbine wakes). We therefore only consider six degrees of freedom: surge  $\zeta$ , heave  $\eta$ , and pitch  $\alpha$  (and their respective velocities) as shown in Figure 1b. See also Figure 2 for more details on the coordinates and platform geometry.

In this model the balance of forces and torques for the surge, heave, and pitch are:

$$E\dot{q} = F,\tag{1}$$

**Figure 1.** Offshore wind turbine floating platform. The platform is a partially submerged, chain-stabilized design (Figure 1 (a) is reproduced from Karimi et al. (2017). Figure 1 (b) is reproduced from Liu and Chertkov (2023)). The turbine parameters correspond to a 5MW design (Papi and Bianchini (2022)).

Figure 2. Offshore wind turbine floating platform, further details and coordinates. (See Betti et al. (2014)).

with the overdot indicating a time, t, derivative. Here  $q = [\zeta, \dot{\zeta}, \eta, \dot{\eta}, \alpha, \dot{\alpha}]^{\top} := [\zeta, u, \eta, v, \alpha, w]^{\top}$ , the inertia matrix E is defined as

$$E = \begin{bmatrix} 1 & 0 & 0 & 0 & 0 & 0 \\ 0 & M_X & 0 & 0 & 0 & M_d \cos \alpha \\ 0 & 0 & 1 & 0 & 0 & 0 \\ 0 & 0 & 0 & M_Y & 0 & M_d \sin \alpha \\ 0 & 0 & 0 & 0 & 1 & 0 \\ 0 & M_d \cos \alpha & 0 & M_d \sin \alpha & 0 & J \end{bmatrix},$$

$$(2)$$

and the forcing vector is

$$F = \begin{bmatrix} u \\ Q_{\zeta} + M_d w^2 \sin \alpha \\ v \\ Q_{\eta} - M_d w^2 \cos \alpha \\ w \\ Q_{\alpha} \end{bmatrix} . \tag{3}$$

The weight forces are:

$$Q_{\zeta}^{(we)} = 0, \tag{4}$$

$$Q_{\eta}^{(we)} = (M_N + M_P + M_S)g,\tag{5}$$

$$Q_{\alpha}^{(we)} = [(M_N dNv + M_P d_{Pv})\sin\alpha + (M_N dNh + M_P d_{Ph})\cos\alpha]$$
 (6)

where  $M_N$ ,  $M_P$  and  $M_S$  are the masses of the systems N, P, and S shown in Figure 2. The  $d_{Pv}$ ,  $d_{Nv}$ ,  $d_{Ph}$ ,  $d_{Nh}$ , are the distances between BS and BP and between BS and BN in the direction parallel to the tower's axis v, and horizontal axis h,  $d_N = \sqrt{d_{Nh}^2 + d_{Nv}^2}$ . Buoyancy forces are defined as:

$$Q_{\zeta}^{b} = 0, \tag{7}$$

$$Q_n^b = -\rho w V_a g, \tag{8}$$

$$Q_{\alpha}^{b} = \rho w V_{g} d_{G} \sin \alpha \tag{9}$$

and wave forces as:

$$Q_{\zeta}^{(wa)} = (\rho V_g + m_x) \cos \alpha \ddot{z},\tag{10}$$

$$Q_{\eta}^{(wa)} = (\rho V_g + m_x) \sin \alpha \ddot{z},\tag{11}$$

$$Q_{\alpha}^{(wa)} = 0$$
 (12)

Wind forces,  $Q^{wi}$ , will be further discussed in Section 4 as we compose them from LES data. We refer the reader to Betti et al. (2014) for the definition of tie rod forces,  $Q^{tr}$ , and hydraulic drag forces,  $Q^{hy}$ . The composite forces are then:

$$Q_{\zeta} = Q_{\zeta}^{(we)} + Q_{\zeta}^{(b)} + Q_{\zeta}^{(wi)} + Q_{\zeta}^{(tr)} + Q_{\zeta}^{(wa)} + Q_{\zeta}^{(hy)}$$

$$Q_{\eta} = Q_{\eta}^{(we)} + Q_{\eta}^{(b)} + Q_{\eta}^{(wi)} + Q_{\eta}^{(tr)} + Q_{\eta}^{(wa)} + Q_{\eta}^{(hy)}$$

$$45 \quad Q_{\alpha} = Q_{\alpha}^{(we)} + Q_{\alpha}^{(b)} + Q_{\alpha}^{(wi)} + Q_{\alpha}^{(tr)} + Q_{\alpha}^{(wa)} + Q_{\alpha}^{(hy)}$$

$$(13)$$

The modifications we made to the platform model in Betti et al. (2014) are that the wind is supplied by a large eddy simulation (LES) of the atmosphere, and the upstream velocity at the hub takes into account the motion of the turbine itself due to wave motions. Furthermore, we use an actuator disk approximation for the turbine, hence, some of the details on how the velocity from the LES calculation enter the platform dynamics will be fully clarified in the subsequent sections. The platform wind drag forcing enters via  $Q^{(wi)}$  as the cell-averaged instantaneous drag. The details of the calculation of the wind drag appear in Section 4. The wave forcing;  $Q^{(wa)}$ ; requires the calculation of  $\ddot{z}$ , the vertical sea elevation acceleration. The ocean waves considered here are specific to a fully developed gravity wave, deep water sea model. See Appendix A for the calculation of the sea surface acceleration.

#### 3 Problem Statement

We aim to determine whether floating motions of a Tension Leg Platform (TLP) affect the *mean* power generated by wind turbines mounted on floating platforms and the wake structure. In this study, we focus on wave motions due to fully developed conditions, which means that we exclude remotely-generated swell and transient, storm-driven conditions. To accomplish this, we will be comparing the steady power output of 5MW turbines, mounted on moving Tension-Leg Platforms (TLPs), to the same turbines mounted on non-moving platforms. Since these turbines are usually arranged into large wind farms composed of many turbines, we also want to determine whether wave motions affect the power generation of a collection of floating platform turbines due to changes in the waking effect of upstream turbines on the other turbines.

We will simplify the dynamics of the turbine at 10m grid spacing by making use of an *actuator disk approximation*. The wind will be generated by a large eddy simulation (LES) approximation of compressible, non-hydrostatic atmospheric flows. The subgrid model uses a one-equation closure to account for unresolved turbulence kinetic energy. The thrust deficit applied by the turbine to the surrounding flow is given by an expression of the form:

$$T = \frac{1}{2}\rho C|\mathbf{u}|\mathbf{u},\tag{14}$$

where  $\rho$  is the atmospheric density, C is the coefficient of thrust, u is the inflow velocity. Further details are presented in Section 4.2.1. Because we are constraining the incoming velocity to be directed normal to the turbine disk, the component of the velocity affecting the thrust, in the absence of platform motions, is:

170 
$$u(t) \cdot \hat{x} = u_{unstream}(t),$$
 (15)

which will be taken at an upstream and on-axis location to the turbine. The computational details of finding  $u_{upstream}$  appear in Appendix B1. When the platform motion is taken into account

$$\boldsymbol{u}(t) \cdot \hat{\boldsymbol{x}} := U(t) = u_{upstream}(t) + u_{platform}(t), \tag{16}$$

where

75 
$$u_{platform}(t) = u(t) + d_P w(t) \cos[\alpha(t)],$$
 (17)

where  $d_P$  is the distance between BS and BP as shown in Figure 2.

# 4 Large Eddy Simulation of the Wind and Turbine Forces

The Large Eddy Simulation model uses compressible, non-hydrostatic, conservation-form equations conserving mass, momenta, and mass-weighted virtual potential temperature. The equations are given in Appendix B (Norman, 2021; Norman et al., 2023b). A semi-discretized, cell-centered, upwind Finite-Volume discretization is used in space with 9th-order-accurate Weighted Essentially Non-Oscillatory (WENO) reconstruction (Liu et al., 1994) with weight mapping (Henrick et al., 2005) for tracers and standard 9th-order Vandermonde-constrained polynomials for density and velocity. A third-order accurate, three-stage Strong Stability Preserving (SSP) Runge-Kutta (RK) is used to discretize in time (see (Gottlieb, 2005)). A CFL value of 0.6 is used in all simulations, considering the total wave speed (acoustics and advection).

The sub-grid-scale (SGS) is modeled with an eddy viscosity model that assumes resolved averages of correlations of unresolved perturbations are proportional to the resolved gradient, the coefficient of proportionality being the eddy viscosity. The eddy viscosity is modeled with a one-equation closure model that actively prognoses SGS unresolved Turbulence Kinetic Energy (TKE) (see (Lilly, 1966, 1967; Smagorinsky, 1963)). TKE is evolved according to shear production, resolved transport, turbulent transport (dissipation), TKE dissipation, and buoyancy modification. At the ocean surface, surface friction is applied using Monin-Obukhov similarity theory (Monin) according to a roughness length determined by the Charnock relation (Mahrt et al., 2003), which estimates roughness length of the ocean surface based on friction velocity,  $u^*$ , in an iterative process involving the following two equations:

$$u^* = \kappa |\mathbf{u}|_{hub} \left( \ln \left( \frac{z_{hub}}{z_0} \right) \right), \tag{18}$$

$$z_0 = -\frac{\alpha}{a} \left( u^* \right)^2,\tag{19}$$

where  $\kappa = 0.4$  is the von Karmann constant,  $\alpha = 0.018$  is the Charnock constant, and |u| is specified for each experiment and enforced as a horizontal average in the turbulent precursor via pressure gradient forcing at hub height.

The model, including all turbine and floating platform motion parameterizations, are coded in portable C++ using the Yet Another Kernel Launcher (YAKL) library (Norman et al., 2023a), a library based on the Kokkos (Trott et al., 2022) portability library, to run on CPUs or Nvidia, AMD, and Intel GPUs.





# 4.1 Turbulent Precursor and Boundary Conditions

The turbine simulations use a mean hub-height velocity of  $\mathbf{u} = (|\mathbf{u}|_{hub}, 0, 0)$ . They are forced with specified inflow at the left x boundary and sponged into the domain over 10% of the x direction domain length. Periodic boundaries are used in the meridional / y direction boundaries, open boundary conditions are used at the right x direction boundary, and no-slip solid wall boundaries are used in the vertical with a Monin-Obukhov friction term at the bottom surface. The specified inflow / outflow boundaries come from a concurrent and horizontally periodic turbulent precursor simulation that simulates an equilibrium turbulent atmospheric boundary layer with pressure gradient forcing that ensures a consistent average horizontal wind speed at turbine hub height. The concurrent turbulent precursor is simulated over the same domain as the turbine simulation, and the pressure gradient force applied to the precursor is also applied to the turbine simulation each time step.

Both the precursor and the turbine simulations use the same surface friction and roughness length throughout. Turbine simulations with and without floating platform motions use identical precursor forcing and pressure gradient forcing. Thus, differences in the fields are *only* due to the floating motions.

#### 4.2 Wind Turbine Parameterization

Wind turbines are parameterized by: (1) applying thrust against the normal horizontal flow passing through the turbine swept plane; and (2) injecting unresolved TKE into the flow to enhance SGS mixing downstream. This parameterization uses a look-up table called a *Reference Wind Turbine* (RWT) that interpolates thrust coefficient  $C_T$ , power coefficient  $C_P$ , and power generation P, as a function of inflow wind speed |U|. The RWT table appears, in figure form, in Figure 3, which we computed with high resolution (2.5m grid spacing) AMR-Wind (Kuhn et al. (2025)) actuator line simulations using an online turbine controller.

# 4.2.1 Actuator Disk Approximation

A one dimensional momentum balance is used to approximate the thrust applied by rotating blades onto a flow (Madsen, 1996).

The thrust is given by:

$$\boldsymbol{T} \cdot \hat{\boldsymbol{x}} = \frac{1}{2} \rho C_T A |\boldsymbol{U}|^2, \tag{20}$$

where  $C_T$  is obtained from the RWT look-up and  $A = \pi \left(D_T/2\right)^2$  is the area of the swept disk.  $D_T$  is the turbine disk diameter. A projection strategy is devised to relate the turbine parametrization to the dynamics on the computational grid. This total disk thrust is divided up according to convex projection weights,  $w_{ijk}$  (described in Appendix B2), to project thrust onto the flow. The indices i, j, k identify the discrete cell volume in the three dimensions, x, y, and z, respectively. Because the Finite-Volume method stores cell-averaged data, this thrust is then also divided by the cell volume  $\Delta x \Delta y \Delta z$  in order to be applicable to cell-averaged quantities. Cell averages are indicated with an overbar. The update that introduces thrust and swirl is defined as:

$$\frac{d\overline{\rho}\overline{u}_{ijk}}{dt} = -\frac{1}{2\Delta x \Delta y \Delta z} C_T \overline{\rho}_{ijk} |U|^2 \cos(\theta_{yaw}) A w_{ijk} - \frac{1}{2\Delta x \Delta y \Delta z} C_Q \overline{\rho}_{ijk} |U|^2 \sin(\theta_{az}) \sin(\theta_{yaw}) A w_{ijk}$$

$$(21)$$

Figure 3. Plot showing the thrust coefficient, power coefficient, and power generation as a function of inflow |U|, in the (5MW) reference wind turbine (RWT) table computed with AMR-Wind. The y-axis units are non-dimensional for thrust and power coefficients and scaled MW for power generation .

$$\frac{d\overline{\rho v}_{ijk}}{dt} = -\frac{1}{2\Delta x \Delta y \Delta z} C_T \overline{\rho}_{ijk} |U|^2 \sin\left(\theta_{yaw}\right) A w_{ijk} + \frac{1}{2\Delta x \Delta y \Delta z} C_Q \overline{\rho}_{ijk} |U|^2 \sin\left(\theta_{az}\right) \cos\left(\theta_{yaw}\right) A w_{ijk},\tag{22}$$

$$\frac{\partial \overline{\rho w}_{ijk}}{\partial t} = \frac{1}{2\Delta x \Delta y \Delta z} C_Q \overline{\rho}_{ijk} |U|^2 \cos(\theta_{az}) A w_{ijk}, \tag{23}$$

where  $\theta_{yaw}$  is the yaw angle of the turbine,  $\theta_{az} = \tan^{-1}{(z/-y)}$  is the azimuth angle with respect to the hub center in reference space averaged over the cell, and  $C_Q = C_P U/(\Omega R_T)$  is the coefficient of torque with  $R_T = D_T/2$  as the turbine radius and  $\Omega$  as the rotation rate in radians per second obtained through a Reference Wind Turbine table interpolation based on inflow speed. In the simulations that follow, the turbines are aligned with the mean flow, and  $\cos(\theta_{yaw}) = 1$ . The portion of the total thrust that does not go into the production of power is injected into unresolved TKE:

$$\frac{d\overline{\rho K}_{ijk}}{dt} = -\frac{1}{2\Delta x \Delta y \Delta z} C_T \overline{\rho}_{ijk} |U|^3 A w_{ijk}, \tag{24}$$

where  $C_{TKE} = \frac{1}{4}(C_T - C_P)$  is the fraction of unused thrust that will contribute to unresolved TKE (Fitch et al., 2012; Bui et al., 2023; Archer et al., 2020). Instantaneous velocities are used to compute the thrust and TKE injection in Eqs (21)-(24). A 30-second average of the inflow velocity is used when interpolating  $C_T^{\star}$ ,  $C_P^{\star}$ , and P from the RWT look-up table (see Figure 3).






#### 245 5 Numerical Results and Discussion

The goal of the numerical simulations that follow is to assess the relative impact of induced inflow velocity perturbations due to floating motions on the power output and wake structure of turbines mounted on floating platforms. The impact is surmised by comparing the power output and wake deficits of turbines affected by floating motions to that of stationary turbines – all other aspects being identical, including the turbulent inflow. The random nature of the waves and turbulent intermittent thrust dictate we report time average power output rather than snapshots in time.

The model used herein uses Cartesian coordinates with equal grid spacing in all directions (10m grid spacing unless otherwise mentioned). The model equations and setup are given in Appendix B. The primary parameter in all of the calculations is the inflow speed. The operational wind speeds for the NREL 5MW turbine span 3-25 m/s. We consider first the power output and wake deficit of a single turbine and then of a simple multiple turbine farm configuration. In the latter case, the turbines' wakes change the power output of waked turbines downstream.

For single-turbine simulations, we simulate with ten different mean hub height precursor wind speeds (5, 7, 9, 11, 13, 15, 17, 19, 21, and 23 m/s) with and without floating motions added to the inflow velocity of the turbine actuator disk for three different turbulent intensities: 25% of default  $(TI_{0.25})$ , 100% of default  $(TI_{1.0})$ , and 200% of default  $(TI_{2.0})$  – all using the Betti et al. (2014) model for floating motions. The default turbulent intensity for a given wind speed is that which naturally arises from a neutral atmospheric boundary layer and the surface roughness length arising from the Charnock relation at a given hub height wind speed. Additionally, the effects of harmonic floating motions (rather than the motions of the Betti et al. (2014) model) are evaluated at  $TI_{1.0}$  for the ten wind speeds with and without floating motions. This leads to 80 total single turbine simulations, each for eight model hours. For the multi-turbine simulations, a grid of 3 rows of 10 turbines (ten turbine diameters apart in both spanwise and streamwise horizontal directions) is simulated with  $TI_{0.25}$  and an inflow average hub height wind speed of 10 m/s for eight model hours. The blockage of subsequent turbines, leads to an average inflow of around 7.5 m/s for the latter nine turbines.

In all simulations in this section, a CFL stability value of CFL = 0.6 is used, and a grid spacing of 10m in each direction is used. Identical turbulent precursor inflow conditions are used for all simulations of a given hub height mean wind speed to ensure the only wake and power differences between simulations with and without floating motions are due solely to floating motions.

For the single turbine we use a  $30 \times 10 \times 6.35$  turbine diameters domain (where D is the turbine diameter) simulated for eight model hours – the last four of which are used for analysis after the neutral ABL has spun up. For multi-turbine simulations, a domain of  $130 \times 40 \times 6.35$  turbine diameters is simulated for eight model hours – the last four of which are used for analysis. Each turbulent precursor is forced with a uniform pressure gradient forcing that penalizes horizontal-mean deviations of hub height wind speed from the intended mean value. The pressure gradient forcing calculated and used in the precursor is also applied to the turbine simulation to ensure pressure gradient advances the flow physically. All simulations contain a dynamical core, an LES closure, and the application of surface friction to impose a roughness length determined by the Charnock relation for the given hub height wind speed. The time step size used for all portions of the model (dynamical core,

LES closure, wind turbine parametrization, surface friction, and floating motion parametrization) is:  $\Delta x (u_{max})^{-1} (CFL) = 10 \text{m} (450 \text{m/s})^{-1} (0.6) = 1.33 \times 10^{-2} \text{s}$ . For single turbine simulations, the turbine is placed at six turbine diameters from the left x direction boundary and in the middle of the domain in the y direction. For multi-turbine simulations, the first column of turbines begins at 11 turbine diameters into the x direction of the domain, and the rows are 10 diameters apart and collectively centered in the y direction.

The neutral boundary layer is initialized with the following potential temperature profile:

$$\theta(z,t=0) = \begin{cases} 300 & \text{if } z < 500\text{m} \\ 300 + 0.08(z - 500) & \text{if } 500 \le z \le 650 \\ 312 + 0.003(z - 650) & \text{if } z > 650 \end{cases}$$
 (25)

The velocity is initialized with a log-law of u velocity targeting the appropriate hub height velocity for the individual experiment with v=w=0. To initiate turbulence, potential temperature perturbations from a random uniform distirbution  $\in [-1,1]$  are added to the lowest 100m of the domain, and horizontal velocities are perturbed with four-period sine functions in transverse directions.

The different turbulent intensities ( $TI_{0.25}$ ,  $TI_{1.0}$ , and  $TI_{2.0}$ ) are created by scaling wind velocity deviations from the mean column from the precursor simulation before applying it as inflow to the turbine simulations by 25%, 100%, and 200%, respectively. These act to mimic turbulent intensities arising from transient stable and unstable boundary layers.

# 5.1 Turbine Power and Velocity - Moving Platform Conditions

In what follows we will refer to outcomes computed with no platform motions as *Fixed*. When the outcomes involve platform motions due to ocean dynamics we will refer to the results as *Moving/Ocean*. A key feature of platform motions affected by a fully-developed (Pierson-Moskowitz) ocean wave spectrum (see Stewart (2008)) is the assumption that the waves have zero-mean random phases. This is in contrast to induced platform motions due to swells created by distant disturbances and storms which tend to have spectra with fixed or slowly changing phases. Consideration of swell conditions is outside of the scope of this study. However, we will explore a simpler case; namely, a sinusoidal perturbation of the incoming wind, with a fixed phase.

The significant wave height is used as the amplitude, and the peak Pierson-Moskowitz frequency is used as the frequency for the sinusoidal forcing – resulting in relatively large amplitude perturbations. We will denote the results of calculations with the oscillating perturbation as *Moving/Oscillating*.

We next describe the computation of U(t) to be used in the actuator disk approximation. For the Fixed conditions case:

$$U(t) = u_{upstream}, (26)$$

and in the Moving/Ocean case:

$$U(t) = u_{upstream} + u_{platform}(t). (27)$$


| $oxed{ oldsymbol{u}_{hub} }$ | $\sigma\left(oldsymbol{u}_{platform} ight)$ |            |             | $\sigma(m{u}_{upstream})$ |            |             |
|------------------------------|---------------------------------------------|------------|-------------|---------------------------|------------|-------------|
|                              | $TI_{2.0}$                                  | $TI_{1.0}$ | $TI_{0.25}$ | $TI_{2.0}$                | $TI_{1.0}$ | $TI_{0.25}$ |
| 5                            | 0.018                                       | 0.010      | 0.003       | 0.267                     | 0.136      | 0.035       |
| 7                            | 0.038                                       | 0.022      | 0.013       | 0.382                     | 0.193      | 0.049       |
| 9                            | 0.058                                       | 0.037      | 0.025       | 0.571                     | 0.289      | 0.073       |
| 11                           | 0.080                                       | 0.057      | 0.042       | 0.673                     | 0.343      | 0.087       |
| 13                           | 0.090                                       | 0.069      | 0.060       | 0.772                     | 0.394      | 0.100       |
| 15                           | 0.105                                       | 0.087      | 0.083       | 1.048                     | 0.536      | 0.138       |
| 17                           | 0.113                                       | 0.101      | 0.101       | 1.149                     | 0.586      | 0.149       |
| 19                           | 0.131                                       | 0.118      | 0.112       | 1.296                     | 0.665      | 0.170       |
| 21                           | 0.132                                       | 0.129      | 0.120       | 1.468                     | 0.751      | 0.192       |
| 23                           | 0.152                                       | 0.133      | 0.133       | 1.668                     | 0.857      | 0.221       |

**Table 1.** Standard deviation of  $u_{platform}$  and  $u_{upstream}$  over a range of hub height wind speeds with different inflow turbulent intensities using Moving/Ocean conditions (see Eq. (27)) for the platform over the last four hours of an eight hour simulations.

The dynamics of  $u_{platform}(t)$  are described by eq:uplatform, with input  $u_{upstream}$  at 10m above the sea surface. The platform motions are described in Section 2. For Moving/Oscillating conditions we have:

$$U(t) = u_{upstream} + A\omega_p \cos(\omega_p t). \tag{28}$$

The amplitude A is determined by the significant height (see Appendix A) and  $\omega_p$  is the peak of the Pierson-Moskowitz spectrum generated by  $u_{upstream}$  at 19.5m above the sea surface.

# 5.2 Velocity and Power, Moving/Ocean Conditions for a Single Turbine

In Table 1 we report the standard deviation of the components of U(t) (see Eqs. (17) and 27) as a function of incoming hub-height wind speed in the precursor inflow, under three inflow turbulent intensities. Platform-induced motions due to ocean and turbulent thrust motions increase as the standard deviation of  $u_{upstream}$  and the hub wind speed increases. In all regimes, the standard deviations of  $u_{platform}$  are much smaller than the standard deviations of  $u_{upstream}$ , meaning the fluctuations due to floating motions are not significant compared to already present turbulent intensity and are, therefore, expected to have relatively minor impact on wake deficit structure and downstream power generation.

Figure 4 plots the ratio of Moving/Ocean x-direction wind speed (disk-normal wind speed) averaged over the hub area divided by that of the Fixed x-direction wind speed averaged over the hub area, averaged over the last four hours of simulations for mean precursor hub height inflow speeds of of 5, 7, 9, and 11 m/s, where power production is below rating. Figure 5 plots are the same except for y-axis instead gives the ratio of power production according to an RWT look-up of |u| as plotted in Figure 4. Figure 6 plots are the same as Figure 4, except for the higher inflow speeds where power is already near or at rating.

In all simulations, the power production of the floating turbines is within 1% of the power production of fixed turbines. At 10 diameters downstream and onward, where another turbine is likely to be placed in a floating offshore farm, the power

Figure 4. Plots of the ratio of averaged horizontal velocity magnitude of Moving/Ocean to Fixed conditions, for hub height wind speeds of 5, 7, 9, and 11 m/s. Averages are integrated over the last four hours of simulation, over the turbine diameter in the y direction, and over the turbine diameter in the z direction. x-axes are x-direction in units of turbine diameter D. y-axes are dimensionless ratios. Note the different y-axis scales in plots.

production is within half a percent, comparing floating to fixed, with the larger differences occurring for lower wind speeds, where the power production is already relatively low.

# 5.3 Motion Effects on Velocities for Moving/Sinusoidal Conditions for a Single Turbine

As a way to assess the extent to which our conclusions regarding floating platform motion effects on power generation depend on the nature of the dynamics of the platform motion, we also simulate cases where we oscillated the hub speed by a sinusoidal (see Eq. (28)). For a given hub wind speed, we use the significant wave height as the amplitude of the sinusoidal and the peak frequency in the sine wave. The outcomes for each wind speed are summarized in Table 2 as well as the bottom right panels of Figures 4, 5, and 6. While the 10D downstream power generation differences due to floating motions remains small (within 0.5-1%), there is a significant effect in the very near wake region of 0-6 turbine diameters downstream due to Moving/Oscillating

Figure 5. Plots of the ratio of averaged power production of Moving/Ocean to Fixed conditions, for hub height wind speeds of 5, 7, 9, and 11 m/s. Averages are integrated over the last four hours of simulation, over the turbine diameter in the y direction, and over the turbine diameter in the z direction. x-axes are x-direction in units of turbine diameters. y-axes are dimensionless ratios. Note the different y-axis scales in plots.

platform motions. As with the floating platform-induced motions, the larger differences are confined to the near-wake region, where a downstream turbine is less likely to be placed in a farm configuration. For all wind speeds, the sinusoidal motion forcing leads to a decrease in power production in the near-wake region (less than 5 turbine diameters downstream). For lower wind speeds, the power production increases due to sinusoidal forcing in the far-wake region; and for larger wind speeds, the power production is decreased due to sinusoidal forcing in the far-wake region (both to modest degrees of less than half a percentage).

# 5.4 Velocity and Power Generation of a Farm of Floating Turbines

The question of whether floating motions have an effect on the power output of a farm of turbines set on floating platforms is considered next. The turbine farm we consider has three rows of turbines, equally spaced and at equal heights, lined up with


Figure 6. Plots of the ratio of averaged horizontal velocity magnitude for Moving/Ocean simulations divided by that of Fixed simulations for hub height wind speeds of 13, 15, 17, 19, 21, and 23 m/s. Averages are integrated over the last four hours of simulation, over the turbine diameter in the y direction, and over the turbine diameter in the z direction. x-axes are x-direction in units of turbine diameters. y-axes are dimensionless ratios. Note the different y- axis scales in plots.

the wind in such a way as to produce the most direct interaction of upstream wakes with downstream turbines. As before, we compare the x-direction velocity at hub height as well as the power produced by turbines in the cluster under Ocean/Moving conditions and Fixed conditions. The Atmosphere conditions are using the  $TI_{1.0}$  (or default neutral ABL) configuration, where floating motions are relatively larger than turbulent fluctuations as compared to the other turbulent intensities. The sea is fully developed, and thus the platform motions due to ocean waves of each turbine are not correlated with each other.

Horizontal wind speed under Moving/Ocean platform conditions averaged over the turbine vertical swept plane ( $z \in [27, 153]$ m) and over  $t \in [4, 8]$ hr is depicted in Figure 7 along with an instantaneous plot at z = 90m, t = 8hr. It is clear that the initial turbine produces the most blockage and turbulence is increased in the streamwise direction as more turbines create additional turbulence down each row. There are also coherent structures forming after the third and fourth turbines that propagate outward from each row in the spanwise direction.

Figure 7. Horizontal velocity magnitude in m/s (a) averaged over  $z \in [27,153]$ m and  $t \in [4,8]$ hr; and (b) at hub height at t = 8hr. Inflow hub wind speed is 10 m/s,  $TI_{1.0}$  atmospheric conditions and platform movement at Moving/Ocean conditions (see Eq. (27)).

| Hub Wind (m/s) | $\sigma\left(oldsymbol{u}_{platform} ight)$ | $\sigma(oldsymbol{u}_{upstream})$ | $\Omega_{peak}$ | $h_{1/3}$ |
|----------------|---------------------------------------------|-----------------------------------|-----------------|-----------|
| 5              | 0.584                                       | 0.136                             | 1.95            | 0.42      |
| 7              | 0.812                                       | 0.193                             | 1.39            | 0.82      |
| 9              | 1.039                                       | 0.289                             | 1.08            | 1.35      |
| 11             | 1.264                                       | 0.343                             | 0.89            | 2.02      |
| 13             | 1.487                                       | 0.394                             | 0.75            | 2.82      |
| 15             | 1.709                                       | 0.536                             | 0.65            | 3.75      |
| 17             | 1.930                                       | 0.586                             | 0.57            | 4.82      |
| 19             | 2.150                                       | 0.665                             | 0.51            | 6.02      |
| 21             | 2.368                                       | 0.751                             | 0.46            | 7.35      |
| 23             | 2.586                                       | 0.857                             | 0.42            | 8.82      |

Table 2. Standard deviation of  $u_{platform}$  and  $u_{upstream}$  over a range of hub height wind speeds in  $TI_{0.25}$  conditions using Moving/Oscillating conditions (see Eq. (28)) for the platform.  $\Omega_{peak}$  is the peak frequency of the Pierson-Moskowitz approximation to a fully developed ocean used as the frequency for the oscillating forcing.  $h_{1/3}$  is the "significant wave height" used as the amplitude for the oscillating forcing.

Figure 8. Plot of time-averaged (between 4 and 8 hours) inflow velocity (m/s) at the ten turbines in the three rows with  $TI_{1.0}$  conditions and Moving/Ocean conditions. The horizontal axis corresponds turbine ID in the columns ordered from from left to right in the x-direction.

For the three rows, Figure 8 depicts the inflow velocity magnitude at each of the turbines. After a significant drop in the speed due to waking of the second turbine, the speed recovers somewhat at turbine 3 and is similar for turbine 3-10.

Figure 9 plots the ratio of the power produced by Moving/Ocean simulation turbines divided by that of Fixed turbines in the three rows averaged over the last four hours of simulation. For this specific farm configuration, we find that the power output

Figure 9. Plot of time-averaged (between 4 and 8 hours) power generation, ratio of Moving/Ocean to Fixed condition simulations.  $TI_{1.0}$  atmospheric conditions. Moving/Ocean simulations.

of the turbine farm is similar between moving and fixed platform simulations. The total farm power output of the 30 turbines differed only by .23% due to floating motions over the last four hours of simulation, though each turbine's power can differ by up to 2% due to floating motions. Due to periodicity in the spanwise boundaries, this simulation depicts infinite rows in the spanwise direction.

The time trace of power is plotted in Figure 10 over the eight hours of simulation for the furthest right turbine in the center row to demonstrate that the simulation has reached an equilibrium state in terms of power production before the  $t \in [4,8]$ hr time averaging.

# 365 6 Conclusions


We used simulations to assess the wake characteristics and power output of a single floating wind turbine as well as that of a cluster of turbines. The goal was to determine whether floating motions would affect wake structure and power output of single or groups of turbines. This assessment was done by comparing the power and wake characteristics of floating platform wind turbines to those of wind turbines that are not subjected to ocean motions.

The simulations consisted of a wind, captured by large eddy simulations (Norman, 2021) of an atmosphere with three inflow turbulence intensities; a mechanical model of a floating chain-stabilized Tension Leg Platform (TLP) due to Betti et al. (2014); and a time dependent empirical model for a fully developed wind waves described by the Pierson-Moskowitz spectrum (see




Figure 10. Plot of time trace of power generation for the furthest right turbine in the center row.  $TI_{1.0}$  atmospheric conditions. Moving/Ocean simulations.

Stewart (2008)). The specific turbine was the NREL 5 MW design. Platforms of this sort are often motion-stabilized, however, we did not take into account any active motion stabilization and control dynamics.

For an isolated floating turbine we found that the platform motions due to a fully developed ocean minimally affected the wake structure downstream. Further, the time-averaged downstream power production was not affected in a significant way by ocean-induced and intermittent thrust-induced platform motions. The floating motion induced perturbations to inflow velocity are quite small compared to already existing turbulent intensity, even at 25% of the naturally occurring neutral boundary layer turbulence, which is the primary reason why observed changes to wake deficits and downstream power generation due solely to floating motions are small.

Figure 11 depicts the spectrum of  $u_{platform}$  and  $u_{upstream}$  for neutral atmospheric conditions and the Moving/Ocean platform configuration, with a hub speed of 11 m/s and fully developed ocean wave conditions. From this figure, we surmise that the spectrum of U(t) is dominated by platform motions in the high frequency regime and by the wind turbulence in the low frequency regime. Furthermore, the most energetic part of the spectrum is associated with wind dynamics and turbulence. The random phase nature of the waves and the dominance of the wind dynamics on U(t) over platform motions suggest that under fully developed ocean conditions neither the single floating turbine or organized clusters of turbines of the type we investigated are affected in a significant way with regard to power generation by ocean motions. Long time averages of the platform acceleration was zero and therefore it was not expected to see, in the average, a net increase or decrease in the power output of a single turbine. However, it was not obvious at the onset that in cluster arrangements the platform dynamics would affect the wake of windward turbines and thus affect the power of the turbines downstream. However, our calculations reveal that


Figure 11. FFT spectra in time of the magnitude of  $u_{platform}(t)$  and  $u_{platform} + u_{inflow}$  as a function of time scale (inverse frequency) at a hub wind speed of 11 m/s with  $TI_{1.0}$  inflow with a smoothing window of 10 samples for clarity. All hub wind speeds and stability profiles share the same local extrema for  $u_{platform}$ .

downwind turbines are not appreciably affected with regard to average power output. Wake disturbances were clearly apparent on turbines subjected to wave motions, however, this was at near distances that are outside of the realm of practicality.

As a check, we compared random ocean perturbations to (zero-mean) sinusoidal fluctuations of the hub speed in order to determine whether random effects in the platform dynamics had a crucial role to play in enhancing/suppressing time-averaged power output of a moving turbine, as compared to pure sinusoidal fluctuations of the hub speed. We found that, for reasonable sinusoidal amplitude and frequency perturbations there was little change on the power output of the turbine in the far-wake, though downstream power production is lowered in the near-wake where downstream turbines are unlikely to be placed.

A full assessment of the effects of wave motions on the power output of turbines mounted on moving platforms necessitates consideration of transient dynamics. Some of the transient cases are obvious: the effect of swells, storms, developing oceans. Some are less obvious. For example, finite-time averages of power output could still be affected by a fully developed ocean: when the hub wind dips up and down around the minimum or maximum wind operating conditions, for example, or when wind directions are not well accounted for by an alignment of the wind and the turbine. Nevertheless, for moored platforms like the TLP design studied here under the most generic of ocean conditions, namely, sustained steady winds and a fully developed

ocean, the power of floating ocean turbines does not appear, in simulations, to be affected significantly, when compared to fixed turbines.

Code availability. The code is available as a fixed DOI located at https://zenodo.org/records/16780228

# Appendix A: Fully Developed Sea Conditions

The fully developed sea surface obeys the Pierson-Moskowitz (PM) spectrum (Stewart, 2008; Pierson and Moskowitz, 1964). For the PM spectrum:  $S_{PM}(\nu) = \frac{0.0081g^2}{(2\pi)^4\nu^5}e^{-\frac{5}{4}(\frac{\nu}{\nu_{peak}})^{-4}}$ , where  $2\pi\nu_{peak} = 0.877g/u_{19.5}$ , and  $u_{19.5}$  is the wind speed at a 19.5m height above the sea surface, g is acceleration,  $\nu$  is frequency. To obtain a random realization, we choose a frequency range of width  $\Delta\nu$ , the amplitude  $a_i$  of the monochromatic wave with angular frequency  $\Omega_i = 2\pi\nu_i$  is  $a_i = \sqrt{2S_{PM}(\nu_i)\Delta\nu_i}$ , where  $\nu_i$  is the central frequency of the interval  $\Delta\nu$ . The sea elevation  $z = \eta(x_p,t)$  is given by:

$$\eta(x_p, t) = \sum_{i=1}^{N} a_i \sin(\Omega_i t - k_i x_p + \epsilon_i), \quad \ddot{\eta} = -\sum_{i=1}^{N} \Omega_i^2 a_i \sin(\Omega_i t - k_i x_p + \epsilon_i), \tag{A1}$$

where  $x_p$  is the x location of the floating platform,  $\epsilon_i$  is a random phase,  $\Omega_i = \sqrt{gk_i}$  for deep water waves, N, the number of spectral components, is chosen at 400. For the wave forcing we require  $u_{19.5}$  to calculate the PM spectrum. This wind speed is taken as the cell average estimate from the large eddy simulation calculation of the wind field, at a height of 19.5 m above the sea surface. The significant wave height, which is used to set A in Eq. (28), is given by:

$$A := H_{1/3} = 0.21 \frac{u_{19.5}^2}{g}. \tag{A2}$$

# **Appendix B: The Large Eddy Simulation Model Equations**

The equations are cast in Cartesian geometry as follows:

$$\partial_t \mathbf{q} + \partial_x \mathbf{f} + \partial_y \mathbf{g} + \partial_z \mathbf{h} = \mathbf{s}$$
 (B1)

$$\mathbf{q} = \begin{bmatrix} \rho \\ \rho u \\ \rho v \\ \rho w \\ \rho \theta \\ \rho q_v \\ \rho K \\ \rho q_\ell \end{bmatrix}, \quad \mathbf{f} = \begin{bmatrix} \rho u \\ \rho u u + p + \tau_{11} \\ \rho u v + \tau_{21} \\ \rho u w + \tau_{31} \\ \rho u \theta + \tau_{\theta 1} \\ \rho u q_v + \tau_{v1} \\ \rho u K + \tau_{K1} \\ \rho u q_\ell + \tau_{\ell 1} \end{bmatrix}, \quad \mathbf{g} = \begin{bmatrix} \rho v \\ \rho v u + \tau_{12} \\ \rho v v + p + \tau_{22} \\ \rho v w + \tau_{32} \\ \rho v \theta + \tau_{\theta 2} \\ \rho v q_v + \tau_{v2} \\ \rho v K + \tau_{K2} \\ \rho v q_\ell + \tau_{\ell 2} \end{bmatrix}$$
(B2)

$$\mathbf{425} \quad \mathbf{h} = \begin{bmatrix} \rho w \\ \rho w u + \tau_{13} \\ \rho w v + \tau_{23} \\ \rho w w + p' + \tau_{33} \\ \rho w \theta + \tau_{\theta 3} \\ \rho w q_v + \tau_{v 3} \\ \rho w K + \tau_{K 3} \\ \rho w q_\ell + \tau_{\ell 3} \end{bmatrix}, \quad \mathbf{s} = \begin{bmatrix} 0 \\ 0 \\ -\rho' g \\ 0 \\ 0 \\ K_S + K_D + K_B \\ 0 \end{bmatrix}$$
(B3)

$$p = (\rho_d R_d + \rho_v R_v) T = C_0 (\rho \theta)^{\gamma}$$
(B4)

where  $\rho$  is density; u, v, and w are wind velocities in the x, y, and z directions, respectively;  $\theta$  is the potential temperature;  $q_v$  is the wet water vapor mixing ratio such that  $\rho_v = \rho q_v$  is the density of water vapor; K is unresolved, sub-grid-scale 430 Turbulent Kinetic Energy (TKE);  $q_\ell$  is a tracer quantity that contributes to mass but not to pressure, p is total pressure;  $C_0 = \left(R_d\left(p_0\right)^{-R_d/c_p}\right)^{\gamma}$  is a constant of proportionality;  $\gamma = c_p/c_v$  is the ratio of specific heats of dry air;  $c_p$  is specific heat of dry air at constant pressure;  $c_v$  is specific heat of dry air at constant volume;  $p_0$  is reference pressure,  $R_d$  is the dry air ideal gas constant,  $R_v$  is the water vapor ideal gas constant;  $K_S$  is turbulent kinetic energy (TKE) shear production;  $K_D$  is TKE dissipation;  $K_B$  is the TKE buoyancy source/sink;  $\tau_{ij} \ \forall i,j \in \{1,2,3\}$  is the unresolved eddy flux of wind velocity;  $\tau_{\theta j} \ \forall j \in \{1,2,3\}$  is eddy flux of potential temperature;  $\tau_{vj}$  is eddy flux of water vapor,  $\tau_{Kj}$  is eddy flux of TKE (i.e., the turbulent transport terms for TKE);  $\tau_{\ell j}$  is the eddy flux of tracers that do not contribute to pressure (e.g., hydrometeors); and g is acceleration due to gravity. Finally, the total density,  $\rho$ , is given by the sum of all mass-contributing densities:  $\rho = \rho_d + \rho_v + \sum_{\ell} \rho_{\ell}$ , where  $\rho_d$  is the density of dry air

The perturbation pressure,  $p' = p - p_H$ , and density,  $\rho' = \rho - \rho_H$ , are perturbations from a dominant hydrostatic balance denoted by:

$$\frac{dp_H}{dz} = -\rho_H g \tag{B5}$$

# The Eddy Fluxes:

The eddy viscosity is considered to be a function of the unresolved TKE and a stability-corrected corrected length scale, giving an eddy flux of:

$$\tau_{ij} = -\rho \left( K_m + \nu \right) \left( \frac{\partial u_i}{\partial x_j} + \frac{\partial u_j}{\partial x_i} - \frac{2}{3} \frac{\partial u_k}{\partial x_k} \delta_{ij} \right) \; ; \; \forall i, j, k \in \{1, 2, 3\}$$
 (B6)

$$\tau_{\theta j} = -\rho \left( \frac{K_m}{Pr_T} + \frac{\nu}{Pr} \right) \frac{\partial \theta}{\partial x_j} \quad ; \quad \forall j \in \{1, 2, 3\}$$
 (B7)

$$\tau_{Kj} = -2\rho (K_m + \nu) \left(\frac{\partial K}{\partial x_j}\right) \; ; \; \forall j \in \{1, 2, 3\}$$
(B8)

$$\tau_{\ell j} = -2\rho \left(\frac{K_m}{Pr_T} + \frac{\nu}{Pr}\right) \left(\frac{\partial \rho_{\ell}}{\partial x_j}\right) \; ; \; \forall j \in \{1, 2, 3\}$$
 (B9)

$$K_m = 0.1L\sqrt{K}$$
 (B10)

$$Pr_T = \frac{\Delta}{1 + 2L} \tag{B11}$$

$$L = \min\left(\frac{0.76\sqrt{K}}{N + \epsilon}, \Delta\right) \tag{B12}$$


$$\Delta = (\Delta x \Delta y \Delta z)^{1/3} \tag{B13}$$

$$N = \begin{cases} \sqrt{(g/\theta)(\partial\theta/\partial z)} & \text{if } \partial\theta/\partial z > 0\\ 0 & \text{otherwise} \end{cases}$$
 (B14)

where  $\epsilon=10^{-20}$  is a small number to avoid division by zero; N is the Brunt-Vaisala frequency, a measure of atmospheric stability; and  $Pr_T$  is the turbulent Prandtl number; Pr is the Prandtl number;  $\nu$  is kinematic viscosity;  $\delta_{ij}$  is the Kronecker delta;  $u_1, u_2$ , and  $u_3$  are the velocity components in the x, y, and z directions (i.e., u, v, and w);  $x_1, x_2$ , and  $x_3$  are the x, y, and z directions; and subset indices that appear only on one side of an equation are an implied sum over available indices (Einstein summation convention).

# 470 TKE Sources and Sinks:

The TKE evolution equation contains advection and turbulent transport (diffusion) on the left-hand-side of equation Eq. (B1). The remaining processes resolved here are the sources and sinks that are, in general, not cast in conservation form. The are defined as:

$$K_S = \rho K_m \sum_{i,j \in \{1,2,3\}} \left( \frac{\partial u_i}{\partial x_j} + \frac{\partial u_j}{\partial x_i} \right) \frac{\partial u_j}{\partial x_i}$$
(B15)

$$K_D = -\rho \left(0.19 + 0.51L\Delta^{-1}\right) \Delta^{-1} K^{3/2} \tag{B16}$$

$$K_B = -\frac{\rho g K_m}{\theta (Pr)} \frac{\partial \theta}{\partial z} \tag{B17}$$


#### **Surface Friction:**

For all surface cells, and for all cells adjacent to an immersed cell, surface friction in each direction is applied via the following flux term added to the eddy fluxes:

$$\tau_{13}^{\star} = -\frac{\kappa^2 u \sqrt{u^2 + v^2}}{\zeta^2 \Delta z} \; ; \; \tau_{23}^{\star} = -\frac{\kappa^2 v \sqrt{u^2 + v^2}}{\zeta^2 \Delta z} \tag{B18}$$


$$\zeta = \ln\left(\frac{\frac{\Delta z}{2} + z_0}{z_0}\right) \tag{B19}$$

where  $z_0$  is the roughness length.

# **B1** Determining the Upstream Velocity

The "upstream" portion of the inflow velocity ("inflow" velocity is defined here as the sum of the upstream velocity normal to the disk and the platform motions normal to the disk) is integrated over a projected disk 2.5 turbine diameters upstream from the turbine base location. In order to do this, first the upstream direction is determined by computing the average horizontal wind velocity  $u_{avg} = (u_{avg}, v_{avg})$  over a cube with sides of length  $D_T$  centered on the turbine hub location. The upstream direction is then computed with  $\theta_{upstream} = \tan^{-1}(v_{avg}/u_{avg})$ . Then, a disk is projected to be centered about the location:

$$\begin{bmatrix} x_{upstream} \\ y_{upstream} \\ z_{upstream} \end{bmatrix} = \begin{bmatrix} x_p - 2D_T \cos(\theta_{upstream}) \\ y_p - 2D_T \sin(\theta_{upstream}) \\ z_p \end{bmatrix}$$
(B20)

using the same procedure in section B2. The integration provides  $u_{upstream}$  and  $v_{upstream}$ , the x and y components of wind velocity integrated over the upstream projected disk. The upstream wind speed normal to the turbine's yaw angle is then determined as  $|\mathbf{u}_{upstream}| = u_{upstream}\cos(\theta_{yaw}) + v_{upstream}\sin(\theta_{yaw})$ .

# **B2** Projecting the Turbine Swept Disk

The turbine swept plane disk is initially projected onto the model grid at the origin facing westward in the y-z plane. It will later be rotated and translated to the hub location and the correct yaw angle. It uses a radially varying thrust shape

function,  $S_T(r_{yz,initial})$ , and a streamwise projection function,  $S_S(x_{initial})$ . Here r is the radial coordinate in the y-z plane:  $r_{yz,initial} = \sqrt{(y_{initial})^2 + (z_{initial})^2}$ , in reference space. The thrust shape function provides degrees of freedom to represent the way thrust increases toward the radial edges of the swept disk (e.g., Figure 13 of Aranake et al. (2015), Figure 6 of Xu et al. (2022), or Figure 11 of Jeong and Ha (2020)). The projection function reduces the discontinuity resulting from thrust application and helps the numerics better resolve the resulting wake. This kind of projection is common for actuator line approaches to turbine parameterizations with Gaussian projections (Martínez-Tossas et al., 2017; Sørensen et al., 2015). This study uses a more compact function for projection but with similar characteristics and for similar reasons.

The disk is initially projected onto the model with a center location of  $(x_0, y_0, z_0) = 0$  and a yaw angle of zero (meaning the turbine swept plane is oriented facing west. The disk is then rotated about the z-axis according to the yaw angle and translated to the appropriate base location and hub height with the following linear operator:

$$\begin{bmatrix} x_p \\ y_p \\ z_p \end{bmatrix} = \begin{bmatrix} \cos(\theta_{yaw}) & -\sin(\theta_{yaw}) & 0 \\ \sin(\theta_{yaw}) & \cos(\theta_{yaw}) & 0 \\ 0 & 0 & 1 \end{bmatrix} \begin{bmatrix} x_{initial} \\ y_{initial} \\ z_{initial} \end{bmatrix} + \begin{bmatrix} x_{base} \\ y_{base} \\ z_{hub} \end{bmatrix}$$
(B21)

The disk projection weights are created by sampling a product of the thrust shaping function and projection function at regular intervals on a Cartesian hexahedron of size  $D_P \times D_T \times D_T$  in the x, y, and z directions where  $D_P$  is the domain of projection (10 cells total) in the x direction for initial points, and  $D_T$  is the turbine swept disk diameter. For each point, the product  $S_T(r_{yz})S_P(x)$  is added to the cell to the cell that contains the point  $(x_p, y_p, z_p)$ . Once all points are sampled, the sum of weights in each cell is computed, and each cell's weight is normalized by the summed total.

The thrust shaping function used here is computed as the piecewise function:

$$S_{T} = \begin{cases} (p_{1}(r))^{a} & \text{if } 0 \leq r 

Once these weights are computed, a weighted sum using these weights provides the integration of a quantity such as wind velocity over the entire disk.

To summarize:



- Over the domain  $[-x_R, x_R] \times [-\frac{D_T}{2}, \frac{D_T}{2}] \times [-\frac{D_T}{2}, \frac{D_T}{2}]$  in the x, y, and z directions, evenly create sample points:  $(x_{initial}, y_{initial}, z_{initial})$
- Initialize all cell weights to zero:  $w_{ijk}^{\star} = 0$
- For each sample point:
  - Compute the rotated and translated location  $(x_p, y_p, z_p)$  for the sample point using ((B21))
  - Locate the cell,  $C_{ijk}$ , in which the point  $(x_p, y_p, z_p)$  is located
  - Atomically add the product of the thrust shape and streamwise projection functions to the weights in cell  $C_{ijk}$ :  $w_{ijk}^{\star} = w_{ijk}^{\star} + S_P\left(x_{initial}\right) S_T\left(r_{yz,initial}\right)$ 
    - "Atomically" is an algorithmic term to denote that when parallelizing over sample points (operating on each simultaneously), the sum and update should be performed in its entirety without any other operation accessing (read or write) cell  $C_{ijk}$ 's memory. This is to avoid data races.
- Compute the sum of  $w_{ijk}^{\star}$  over all cells
- Divide each weight by the total sum:  $w_{ijk} = w_{ijk}^{\star} / \sum_{ijk} w_{ijk}^{\star}$
- For any set of cell averages of a quantity,  $\overline{q}_{ijk}$ , compute the disk-integration of that quantity with the sum

$$q_{disk} = \sum_{ijk} \overline{q}_{ijk} w_{ijk}$$
 (B24)

In all simulations in this paper, a CFL value of 0.6 is used.

#### **B3 Neutral AWAKEN Experiment**

The Large Eddy Simulation code has been validated against standard neutral and convective boundary layer results, and here, it is validated against the AMR-Wind (Kuhn et al., 2025; Brazell et al., 2021) results for a modestly convective boundary layer test case for both turbulent boundary layer results and turbine wake results for the NREL 5MW turbine used in this study. To initialize, potential temperatures are linearly interpolated between values of 300K, 300K, 308K, and 311.45K at heights of 0m, 750m, 850m, and 2000m, respectively. The u and v winds are initialized with a log law setting u = 9.873m/s and v = 5.7m/s at hub height (90m), and vertical velocity is initialized to zero. AMR-Wind uses incompressible equations with a Boussinesque buoyancy term with a density of one kg per cubic meter throughout; and portUrb uses non-hydrostatic, compressible equations with the surface pressure set to enforce a surface density of one kg per cubic meter at a surface temperature of 300K. Both




models perturbe the lowest 50m of the domain with random potential temperature perturbations and use transverse four-period sine waves for u and v velocity in the lowest 100m.

Coriolis is active with a latitude of  $40^{\circ}$ N. Surface roughness is set to  $z_0 = 0.01$ m. The domain is  $5.12 \times 5.12 \times 1.92$  km in the x, y, and z directions, respectively. A uniform grid spacing of 10m is used for portUrb, while AMR-Wind has multiple grids that reach 2.5m grid spacing around the turbine with 10m grid spacing at the coaresest level. Both models use pressure gradient forcing to force a hub-height horizontal average wind speed of (9.873, 5.7)m/s for the horizontal velocity vector. That pressure gradient forcing is calculated in the turbulent precursor and then used in both the precursor and primary turbine simulation. The precursor and turbine simulations are run for 20,000 model seconds, the first 15,000s used as spin-up and the last 5,000s as the performance period. The average pressure gradient forcing over  $t \in [14000, 15000]$ s is used throughout the performance period in the main turbine simulation. The turbine is placed at 1,800m into the domain in the x and y directions. A small surface heat flux of 0.005 K m / s is added to the surface to create a modestly convective boundary layer.

Figure B1 plots the domain-averaged column of horizontal velocity, potential temperature, and turbulence intensity averaged in time over  $t \in [15000, 20000]$ s for AMR-Wind and portUrb. The profiles are quite similar with portUrb carrying bit more wind speed and turbulence intensity throughout the boundary layer. Figure B2 plots resolved turbulent velocity correlations averaged over  $t \in [15000, 20000]$ s for AMR-Wind and portUrb. The magnitudes and shapes of the profiles are very similar with portUrb exhibiting some oscillations at the surface peak and at times larger features at the top of the boundary layer. Figure B3 plots resolved turbulent temperature-velocity correlations averaged over the same time range. The  $u'\theta'$  and  $v'\theta'$  profiles are similar with portUrb exhibiting stronger features at the top of the boundary layer. The  $w'\theta'$  profile shows more differences, which is somewhat unsurprising, given the large differences in the buoyancy term. portUrb uses a non-hydrostatic and compressible discretization, where temperature perturbations propagate into density perturbations through acoustics, and the buoyancy term uses density perturbations to determine buoyant forcing. AMR-Wind is incompressible and uses a fixed Boussinesque buoyancy term based to deviations from background temperature (fixed at  $T_0 = 300K$  throughout the simulation).

Figure B4 plots the turbine wakes at hub height (z=90m) as a function of y location averaged over  $t \in [15000, 20000]$  for portUrb and AMR-Wind at 2, 4, 6, and 8 turbine diameters downstream of the turbine – along the direction of the wake, which is  $30^{\circ}$ . Both exhibit very similar features and magnitudes, though portUrb is running at  $4\times$  larger grid spacing. Both show the asymmetrical profile due to swirl, and both show very similar wake recovery downstream. There are a bit larger differences in Figure B5, which plots wakes at the same distances downstream as a function of z location at the hub center, following the wake direction of  $30^{\circ}$ . The wakes are quite similar higher up, but nearer to the surface, portUrb has a stronger wake deficit – likely due to the coarser grid spacing leading to surface friction to be less separated from the turbine wake.

In summary, despite being formulated quite differently as a model, portUrb replicates the salient features of both the boundary layer turbulence and the turbine wake deficit shape and magnitude compared to AMR-Wind.

Figure B1. Domain-averaged profiles over  $t \in [15000, 20000]$ s. TI stands for turbulence intensity.  $z_i = 800$ m is the boundary layer height.

Author contributions. Authors contributed equally to the research and its reporting.

Competing interests. The Authors report no competing interests.

Acknowledgements. This research used resources of the Oak Ridge Leadership Computing Facility at the Oak Ridge National Laboratory, which is supported by the Office of Science of the U.S. Department of Energy under Contract No. DE-AC05-00OR22725. This research uses an ASCR Advanced Computing Challenge (ALCC) award titled "High-fidelity modeling and next-generation surrogate models for floating offshore wind energy."

This material is based upon work supported by the U.S. Department of Energy, Office of Science, Offices of Advanced Scientific Computing Research and Biological and Environmental Research through the FLOWMAS Energy Earthshot Research Center.

Figure B2. Resolved turbulent velocity correlations. Perturbations are computed as the deviation from the mean column, and correlations are averaged over multiple snapshots within  $t \in [15000, 20000]$ s.  $z_i = 800$ m is the boundary layer height.

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

Figure B3. Resolved turbulent temperature correlations. Perturbations are computed as the deviation from the mean column, and correlations are averaged over multiple snapshots within  $t \in [15000, 20000]$ s.  $z_i = 800$ m is the boundary layer height.

Figure B4. Hub height (z = 90m) horizontal velocity divided by inflow wind speed (11.4 m/s) in the y-direction at 2, 4, 6, and 8 turbine diameters downstream of the turbine for  $t \in [15000, 20000]$ s along the wake direction (30°).

Figure B5. Horizontal velocity divided by inflow wind speed (11.4 m/s) in the z-direction at the turbine center x = 1800m at 2, 4, 6, and 8 turbine diameters downstream of the turbine for  $t \in [15000, 20000]$ s along the wake direction (30°).

- Michael Brazell, Shreyas Ananthan, Ganesh Vijayakumar, Lawrence Cheung, Michael Sprague, ExaWind Exascale Computing Project

  Team, High Fidelity Modeling Project Team, et al. Amr-wind: adaptive mesh-refinement for atmospheric-boundary-layer wind energy simulations. In APS Division of Fluid Dynamics Meeting Abstracts, pages T29–007, 2021.
  - K. Brown, G. Yalla, L. Cheung, J. Frederik, D. Houck, N. deVelder, E. Simley, and P. Fleming. Comparison of wind-farm control strategies under realistic offshore wind conditions: wake quantities of interest. Wind Energy Science, 10(8):1737–1762, 2025. https://doi.org/10.5194/wes-10-1737-2025. URL https://wes.copernicus.org/articles/10/1737/2025/.
- Hai Bui, Mostafa Bakhoday-Paskyabi, and Mohammadreza Mohammadpour-Penchah. Implementation of a simple actuator disc for large eddy simulation (sadles-v1. 0) in the weather research and forecasting model (v4. 3.1) for wind turbine wake simulation. *EGUsphere*, 2023:1–24, 2023.
  - Lawrence Cheung, Myra L Blaylock, Kenneth Brown, Nathaniel deVelder, Thomas G Herges, Alan Hsieh, David C Maniaci, and James Cutler. Comparison of simulated and measured wake behavior in stable and neutral atmospheric conditions. In *AIAA SCITECH 2022 Forum*, page 1923, 2022.
  - Lawrence C. Cheung, Kenneth A. Brown, Daniel R. Houck, and Nathaniel B. deVelder. Fluid-dynamic mechanisms underlying wind turbine wake control with strouhal-timed actuation. *Energies*, 17(4), 2024. ISSN 1996-1073. https://doi.org/10.3390/en17040865. URL https://www.mdpi.com/1996-1073/17/4/865.

635

- S Cioni, F Papi, L Pagamonci, A Bianchini, N Ramos-García, G Pirrung, R Corniglion, A Lovera, J Galván, R Boisard, et al. Branlard, 710 e., jonkman, j., and robertson, a.: On the characteristics of the wake of a wind turbine undergoing large motions caused by a floating structure: an insight based on experiments and multi-fidelity simulations from the oc6 project phase iii. *Wind Energ. Sci*, 8:1659–1691.
  - Hannah Darling, David P. Schmidt, Shengbai Xie, Jasim Sadique, Arjen Koop, Lu Wang, Will Wiley, Roger Bergua Archeli, Amy Robertson, and Thanh Toan Tran. Oc6 phase iv: Validation of cfd models for stiesdal tetraspar floating offshore wind platform. *Wind Energy*, 28(1):e2966, 2025. https://doi.org/https://doi.org/10.1002/we.2966. URL https://onlinelibrary.wiley.com/doi/abs/10.1002/we.2966. e2966 we.2966.
  - Anna C Fitch, Joseph B Olson, Julie K Lundquist, Jimy Dudhia, Alok K Gupta, John Michalakes, and Idar Barstad. Local and mesoscale impacts of wind farms as parameterized in a mesoscale nwp model. *Monthly Weather Review*, 140(9):3017–3038, 2012.
  - A. Fontanella, A. Fusetti, S. Cioni, F. Papi, S. Muggiasca, G. Persico, V. Dossena, A. Bianchini, and M. Belloli. Wake development in floating wind turbines: new insights and an open dataset from wind tunnel experiments. *Wind Energy Science*, 10(7):1369–1387, 2025. https://doi.org/10.5194/wes-10-1369-2025. URL https://wes.copernicus.org/articles/10/1369/2025/.
  - J. A. Frederik, E. Simley, K. A. Brown, G. R. Yalla, L. C. Cheung, and P. A. Fleming. Comparison of wind farm control strategies under realistic offshore wind conditions: turbine quantities of interest. Wind Energy Science, 10(4):755–777, 2025. https://doi.org/10.5194/wes-10-755-2025. URL https://wes.copernicus.org/articles/10/755/2025/.
- Sigal Gottlieb. On high order strong stability preserving runge-kutta and multi step time discretizations. *Journal of scientific computing*, 25: 105–128, 2005.
  - Andrew K Henrick, Tariq D Aslam, and Joseph M Powers. Mapped weighted essentially non-oscillatory schemes: achieving optimal order near critical points. *Journal of Computational Physics*, 207(2):542–567, 2005.
  - Jae-Ho Jeong and Kwangtae Ha. Numerical investigation of three-dimensional and vortical flow phenomena to enhance the power performance of a wind turbine blade. *Applied Sciences*, 11(1):72, 2020.
- 645 H. M. Johlas, L.A. Martinez-Tossas, D. P. Schmidt, M.A. Lackner, and M.J. Churchfield. Large eddy simulations of floating offshore wind turbine wakes with coupled platform motion. *IOP Conference Series, Journal of Physics*, 1256:012018, 2019.
  - H. M. Johlas, L.A. Martinez-Tossas, D. P. Schmidt, M.A. Lackner, and M.J. Churchfield. Large eddy simulations of oshore wind turbine wakes for two floating platform types. *IOP Conference Series, Journal of Physics*, 1452:012034, 2020.
  - Meysam Karimi, Matthew Hall, Brad Buckham, and Curran Crawford. A multi-objective design optimization approach for floating offshore wind turbine support structures. *Journal of Ocean Engineering and Marine Energy*, 3:69–87, 2017.
  - Michael B Kuhn, Marc T Henry de Frahan, Prakash Mohan, Georgios Deskos, Matt Churchfield, Lawrence Cheung, Ashesh Sharma, Ann Almgren, Shreyas Ananthan, Michael J Brazell, et al. Amr-wind: A performance-portable, high-fidelity flow solver for wind farm simulations. *Wind Energy*, 28(5):e70010, 2025.
- Valerie-M Kumer, Joachim Reuder, Manfred Dorninger, Rudolf Zauner, and Vanda Grubišić. Turbulent kinetic energy estimates from profiling wind lidar measurements and their potential for wind energy applications. *Renewable Energy*, 99:898–910, 2016.
  - Ryan Kyle, Yeaw Chu Lee, and Wolf-Gerrit FrÃ!/4h. Propeller and vortex ring state for floating offshore wind turbines during surge. *Renewable Energy*, 155:645–657, 2020. ISSN 0960-1481. https://doi.org/https://doi.org/10.1016/j.renene.2020.03.105. URL https://www.sciencedirect.com/science/article/pii/S0960148120304377.
- Frank Lemmer, Wei Yu, and Po Wen Cheng. Iterative frequency-domain response of floating offshore wind turbines with parametric drag.

  Journal of Marine Science and Engineering, 6(4), 2018. ISSN 2077-1312. https://doi.org/10.3390/jmse6040118. URL https://www.mdpi. com/2077-1312/6/4/118.

- YuanTso Li, Wei Yu, and Hamid Sarlak. Wake structures and performance of wind turbine rotor with harmonic surging motions under laminar and turbulent inflows. *Wind Energy*, page e2949, 2024.
- Zhaobin Li, Guodan Dong, and Xiaolei Yang. Onset of wake meandering for a floating offshore wind turbine under side-to-side motion.

  Journal of Fluid Mechanics, 934:A29, 2022.
  - Douglas K Lilly. The representation of small-scale turbulence in numerical simulation experiments. In *Proc. IBM Sci. Comput. Symp. on Environmental Science*, pages 195–210, 1967.
  - Douglas Keith Lilly. On the application of the eddy viscosity concept in the inertial sub-range of turbulence. *NCAR manuscript*, 123, 1966. Xu-Dong Liu, Stanley Osher, and Tony Chan. Weighted essentially non-oscillatory schemes. *Journal of computational physics*, 115(1):
- 670 200–212, 1994.
  - Yihan Liu and Michael Chertkov. Deciphering the dance of the winds and waves: Unraveling anomalous dynamics in floating offshore wind turbines. *arXiv* preprint *arXiv*:2402.00018, 2023.
  - Helge Aagaard Madsen. A cfd analysis of the actuator disc flow compared with momentum theory results. In *Proceedings of the 10th Symposium on Aerodynamics of Wind Turbines*, pages 109–124. IEA Joint Action, 1996.
- 675 Larry Mahrt, Dean Vickers, Paul Frederickson, Ken Davidson, and Ann-Sofi Smedman. Sea-surface aerodynamic roughness. *Journal of Geophysical Research: Oceans*, 108(C6), 2003.
  - Luis Martinez, Stefano Leonardi, Matthew Churchfield, and Patrick Moriarty. A comparison of actuator disk and actuator line wind turbine models and best practices for their use. In 50th AIAA Aerospace Sciences Meeting including the New Horizons Forum and Aerospace Exposition, page 900, 2012.
- Luis A Martínez-Tossas, Matthew J Churchfield, and Charles Meneveau. Optimal smoothing length scale for actuator line models of wind turbine blades based on gaussian body force distribution. *Wind Energy*, 20(6):1083–1096, 2017.
  - J. C. McWilliams and J. M. Restrepo. The wave-driven ocean circulation. Journal of Physical Oceanography, 29:2523–2540, 1999.
  - J. C. McWilliams, J. M. Restrepo, and E. M. Lane. An asymptotic theory for the interaction of waves and currents in coastal waters. *Journal of Fluid Mechanics*, 511:135–178, 2004.
- Thomas Messmer, Joachim Peinke, Alessandro Croce, and Michael Hölling. The role of motion-excited coherent structures in improved wake recovery of a floating wind turbine. *Journal of Fluid Mechanics*, 1018:A23, 2025.
  - Aaron Miller, Byungik Chang, Roy Issa, and Gerald Chen. Review of computer-aided numerical simulation in wind energy. *Renewable and sustainable energy Reviews*, 25:122–134, 2013.
  - Andreĭ Monin. Basic laws of turbulent mixing in the surface layer of the atmosphere.
- 690 Matthew Norman, Isaac Lyngaas, Abhishek Bagusetty, and Mark Berrill. Portable c++ code that can look and feel like fortran code with yet another kernel launcher (yakl). *International Journal of Parallel Programming*, 51(4):209–230, 2023a.
  - Matthew R Norman. A high-order weno-limited finite-volume algorithm for atmospheric flow using the ader-differential transform time discretization. *Quarterly Journal of the Royal Meteorological Society*, 147(736):1661–1690, 2021.
- Matthew R Norman, Christopher Eldred, and Muralikrishnan Gopalakrishnan Meena. Investigating inherent numerical stabilization for the moist, compressible, non-hydrostatic euler equations on collocated grids. *Journal of Advances in Modeling Earth Systems*, 15(10): e2023MS003732, 2023b.
  - Yang Pan and Cristina L Archer. A hybrid wind-farm parametrization for mesoscale and climate models. *Boundary-Layer Meteorology*, 168:469–495, 2018.

© Author(s) 2025. CC BY 4.0 License.

- F Papi and A Bianchini. Technical challenges in floating offshore wind turbine upscaling: A critical analysis based on the nrel 5 mw and iea

  15 mw reference turbines. *Renewable and Sustainable Energy Reviews*, 162:112489, 2022.
  - W.J. Pierson and L. Moskowitz. A proposed spectral form for fully developed wind seas based on the similarity theory of s. a. kitaigorodskii. *Journal of Geophysical Research*, 69:5181–5190, 1964.
  - J. M. Restrepo. Wave breaking dissipation in the wave-driven ocean circulation. Journal of Physical Oceanography, 37:1749–1763, 2007.
- A. F. P. Ribeiro, D. Casalino, and C. S. Ferreira. Nonlinear inviscid aerodynamics of a wind turbine rotor in surge, sway, and yaw motions using a free-wake panel method. *Wind Energy Science*, 8(4):661–675, 2023. https://doi.org/10.5194/wes-8-661-2023. URL https://wes.copernicus.org/articles/8/661/2023/.
  - Thomas Sebastian and Matthew Lackner. Offshore floating wind turbines-an aerodynamic perspective. In 49th AIAA aerospace sciences meeting including the new horizons forum and aerospace exposition, page 720, 2011.
- Joseph Smagorinsky. General circulation experiments with the primitive equations: I. the basic experiment. *Monthly weather review*, 91(3): 99–164, 1963.
  - Jens N Sørensen, Robert F Mikkelsen, Dan S Henningson, Stefan Ivanell, Sasan Sarmast, and Søren J Andersen. Simulation of wind turbine wakes using the actuator line technique. *Philosophical Transactions of the Royal Society A: Mathematical, Physical and Engineering Sciences*, 373(2035):20140071, 2015.
  - R. H. Stewart. Introduction to Physical Oceanography. Texas A&M University, 2008.
- Christian R. Trott, Damien Lebrun-Grandié, Daniel Arndt, Jan Ciesko, Vinh Dang, Nathan Ellingwood, Rahulkumar Gayatri, Evan Harvey, Daisy S. Hollman, Dan Ibanez, Nevin Liber, Jonathan Madsen, Jeff Miles, David Poliakoff, Amy Powell, Sivasankaran Rajamanickam, Mikael Simberg, Dan Sunderland, Bruno Turcksin, and Jeremiah Wilke. Kokkos 3: Programming model extensions for the exascale era. *IEEE Transactions on Parallel and Distributed Systems*, 33(4):805–817, 2022. https://doi.org/10.1109/TPDS.2021.3097283.
  - L. Wang, A. Robertson, J. Jonkman, and Y.H. Yu. OC6 phase I: Improvements to the OpenFAST predictions of nonlinear, low-frequency responses of a oating offshore wind turbine platform. *Renewable Energy*, 187:282–301, 2022.
  - Nathaniel J. Wei and John O. Dabiri. Phase-averaged dynamics of a periodically surging wind turbine. *Journal of Renewable and Sustainable Energy*, 14(1):013305, 02 2022. ISSN 1941-7012. https://doi.org/10.1063/5.0076029. URL https://doi.org/10.1063/5.0076029.
  - B. Wena, X. Donga, X. Tian, Z. Penga, W. Zhanga, and K. Wei. The power performance of an offshore floating wind turbine in platform pitching motion. *Energy*, 154:508–521, 2018.
- Yu-Ting Wu and Fernando Porté-Agel. Modeling turbine wakes and power losses within a wind farm using les: An application to the horns rev offshore wind farm. *Renewable Energy*, 75:945–955, 2015.
  - Ziyi Xu, Min Chang, Junqiang Bai, and Bo Wang. Computational investigation of blade–vortex interaction of coaxial rotors for evtol vehicles. *Energies*, 15(20):7761, 2022.
- H. Yang, M. Gea, M. Abkar, and X.I.A. Yang. Large-eddy simulation study of wind turbine array above swell sea. *Energy*, 256:124674, 2022.