# Peer review of "Power Output and Downstream Wake Modifications of Floating Turbines Subjected to Ocean Motions Using a Tension-Leg Platform"

_Wind Energy Science, 2025_

## Referee Comment (RC1)

The considered manuscript describes a simulation study on the wake of a 5MW TLP FOWT using an LES solver. The turbine is included in the simulations with an actuator disk, while the tower top motions are computed from an analytical model. All simulations were performed in a highly turbulent inflow. From comparisons between the fixed case and cases with platform motion it is concluded that the wake deficit and turbine power is not (or only barely) altered by the platform motions.

From my point of view, the topic of the paper is timely and relevant to the readership of WES. The tools used and the approach seem suitable to address the problem. However, the considered manuscript has a number of fundamental weaknesses and ambiguities. One of the weaknesses is that there is lack of clear definitions and referencing, which forced me to interpret some facts. Please correct me in case I interpreted some of them in the wrong way. The most important issues from my point of view are the following:

- The simulations done are not properly defined in terms of the choice of environmental parameters, origin of turbine and floating platform parameters and simulation settings.
  - To my current understanding (see below), the turbine parameters (mostly Ct) are not realistic (Ct of 1 below rated).
  - In case the common definition of the turbulence intensity (e.g. the one used in **IEC** 61400-1) is used, extremely high and far from typical turbulence intensities were used (at least to my current knowledge).
  - The model and hydrodynamic parameters of the TLP are not properly referenced.
- A verification of the simulation method and setup is missing. Concurrently, the presented results do not allow to judge on the plausibility of the simulations. It is not clear whether the wake in the simulation of the fixed case really shows a realistic behaviour.
- The scope of work and literature review are written in a quite confusing way.
- The paper is generally difficult to understand, since a red line is not easy to find and a large number of vague terms (e.g. floating platform effects, floating turbine dynamics, quasi standard distribution induced motions, quasi standard distribution induced motions, ocean dynamic, turbulent thrust intermittency, dynamic coupling of turbine/atmosphere, buoyantly floating moored designs,….) are used.
- The results are not set into a context with comparable works.
- Scales of some graphs (e.g. Figure 6) most likely lie within the accuracy of the model (differences on a scale from 99.992% to 100.002%).
- The main conclusion – the floating motion is not relevant for power production and wake development – might be dependent on the choice of parameters (mainly Ct of 1 and TI 25% and much higher), which are questionable in my eyes (according to my current understanding/interpretation). This needs to be clarified before a judgement on the results is possible.

All in all, the paper would need to be considerably reworked to be acceptable. In case the Ct and TI values really turn out to be unrealistic, the simulations will most likely have to be repeated completely and reanalysed. I would recommend the authors to take their time and revise the simulations and paper. I would be willing to review the paper again if it is considerably reworked. However, it is up to the authors to decide whether they would better like to withdraw it and submit it again, when the simulations and parameters are refined.

Due to the major concerns I encountered while reading, I stopped the in depth commenting after section 4. See the comments for the parts before this below:

Introduction: The introduction tries to explain the context of the work done in a rather broad way (e.g. explanation of stochastic ocean waves, modelling of FOWTs in general, ...). On the one hand, this approach is ambitious and commendable. However, on the other hand it does not really end up

in a red line but just touches many fields without clear conclusions. In addition, it seems that the introduction of previous literature and explanation of the approach of this study is mixed up quite a lot. It is therefore very hard to follow the text. I strongly recommend to reorder the introduction (e.g. Motivation; basics, previous works in the field, presentation and delimitation of the own approach). In addition, there are numerous sentences that I either could not understand or that lack precision. Therefore, the introduction has to be considerably reworked to be understandable.

Turbine: You seem to use the NREL 5MW wind turbine. Is this correct? Of yes: This is not clearly stated. Instead, it is stated: *The turbine parameters correspond to a 5MW design (Papi and Bianchini (2022)). However, there is no turbine defined in the paper. Instead, the NREL 5MW is used.* In case you also use the NREL 5MW, it is mandatory to cite the original source of this design.

Platform dynamics: The section starts with the definition of DOFs for the turbine modelling. However, it was not clear to me at that point in which way the turbine should be modelled. It seems that an analytical model is used. Please state this clearly in the introduction, where you discuss modelling approaches and define the scope of the work.
It seems that parts to the section reproduce selected parts of the work of Betti et al. From my point of view, the selection seems a bit random, while no complete picture of how turbine motions are computed from wave elevations and wind excitation is drawn. It would be more convenient to explain the model in general, while referring to Betti et al. for details and highlighting how the changes of the authors are integrated into the model.

Problem statement: It is not defined, which TLP model is used in the study. In this section, a general idea of the paper is shortly described. This would suit better at the beginning of the paper. In addition, a very brief introduction and formulas redarguing the AD are given. Placing this information here seems to be a bit random, since modelling details are given in the following sections step by step.

Turbulent Precursor and Boundary Conditions: It is stated an atmospheric boundary layer is simulated. However, to my knowledge such flow situation is pretty complex and needs to be defined in some way. This is not done here.

Wind Turbine Parametrization: It is stated that thrust and power of the wind turbine are calculated by own actuator line simulations. However, no details on the simulations are given. No verification of the results is given. In addition, there is no reference given for the operational parameters of the wind turbine. The thrust and power coefficients given in Figure 3 do no seem to be realistic: 1. The nominal power of the 5MW turbine is reached at around 14 m/s. This is very uncommon for a typical Offshore Wind Turbine. The thrust coefficient (1.0) is far from optimal operation at below rated wind (should be somewhere between 0.8 and 0.9). The thrust coefficient does not seem to change continuously between 7 and 12.5 m/s. This hints to a malfunction of the rotor speed control. The controller is not described or referenced.
At my current state of understanding, I believe that the simulations are only used to generate a Cp and Ct look-up table, which is then used by the AD model. If this is the case, I strongly recommend to use look-up tables from literature for e.g. the NREL 5MW wind turbine, since the direct simulation of these quantities requires a lot of effort, is prone to errors and would need to be described in a more detailed way.

Numerical Results and Discussion: The choice of the distribution of significant wave height and wind speed is not justified. It has to be made clear that this is a specific choice, which represents maybe a class of sites or similar. Table 2 should be shown at the beginning of the section and should contain units. The choice of the TI seems very surprising to me. Turbulence intensities of 25%, 100% and 200% are chosen. In case the common definition of TI is used (e.g. the one used in **IEC**

61400-1), this is much higher than what could be expected in typical offshore conditions (something between 2 and 20%).

Results:
- There seem to be numerical issues in the 5m/s case in Figure 5.
- It seems that the scale in Figure 6 is extremely zoomed in and might go far beyond the accuracy of the method.

Comments on sentences:

Line 2: *waking* - I am not aware of this term in the context of Floating Wind.
Line 6: *The ocean dynamics enter as fully developed waves derived from the Pierson-Moskowitz spectrum.*- Is this information so relevant that it needs to be mentioned in the abstract?
Line: 6-8 – This sentence should be reformulated to be more clear.
Line 9: *coherent and large amplitude harmonic floating* – What is this? Please clarify
Line 26: *An ocean platform would be exposed to a variety of different waves.*- Please clarify the meaning of ocean platform. Do you mean floating platform? Please be consistent in the naming.

Line 50 ff. *LES has been used to compare the effect of moving platform turbine wakes to those of a fixed platform.*- Please clarify this sentence.

Line 58 ff. *For instance, in floating turbines with a surge motion frequency commiserate with the rotor speed, a vortex ring state can be encountered if the blade tips interact with the vortex emerging from the previous blade (Sebastian and Lackner, 2011; Kyle et al., 2020), and consequently change the near field wake.*From my understanding, these are two different things: 1. there is an ongoing discussion if it is really realistic that a vortex ring state may occur. Actually, some researchers argue convincingly that this would not be the case. Instead, a blade-wake-interaction may occur. (see F. Papi, J. Jonkman, A. Robertson, A. Bianchini, Going beyond BEM with BEM: an insight into dynamic inflow effects on floating wind turbines, Wind Energ Sci 9 (2024) 1069–1088). 2. I am not aware of the role of the rotor speed in this context and why it has to commiserate with the surge motion frequency. Please clarify this (ideally with a source).

Line 71 ff.*In the context of LES for mesoscale models, approaches include using LES data to build models for hub height wind speeds and other relevant model parameters Pan and Archer (2018) for static turbine configurations.*- Please clarify this sentence.

Line 78 ff. *These model and simulation efforts are aimed toward discerning first order dynamic/structural issues that will inform the design and construction of these FOWTs. For example, Johlas et al. (2020) compare wake characteristics of turbines mounted on different floating platforms, and their waks to those of fixed turbines.* - Please clarify these sentences. I do not understand the first sentence and cannot find a connection to the second one. What do you mean by *dynamic/structural issues?*

Line 81 ff. *To model floating platform effects initially at an intermediate resolution using 10m grid spacing to better understand the dynamics relevant to mesoscale, our approach is to first extend LES simulations to include floating turbine dynamics such that power production and subsequently turbulent kinetic energy (i.e. the quantities of interest in a mesoscale parameterization), are directly modified by the turbine motions.* - Please reorder and simplify this sentence.

Line 87 ff. *This approach allows us to understand how floating platform motions from*

*ocean dynamics and turbulent thrust intermittency alters key elements of the flow used in mesoscale parameterizations including hub-height wind speeds and directions, turbulence, and turbine power output over the time scales present in a mesoscale simulation.* - Please make two sentences of this and clarify the meaning.

*Line 90 ff. Our approach distinguishes itself from the existing literature in its breadth, by the dynamic coupling of turbine/atmosphere and turbines, consideration of different atmospheric turbulent conditions, and consideration of single turbines as well
as simple cluster arrangements of multiple turbines.*- This seems to be a rather important statement. Unfortunately, I do not really understand what is exactly meant. Therefore, I cannot really judge on the content. Please try to make this more clear so that we can discuss the content of the sentence.

Line 93-106: I do not see the connection between these three subsequent paragraphs. What is the red line of the argumentation here? The sentence *"Recently there has been a dearth of papers that concern the coupled dynamics of floating turbines and the atmospheric
boundary layer that they occupy or are planned to occupy."*seems to stand for its own without any context. In addition, I do not really understand the meaning: What does "papers that concern the coupled dynamics of floating turbines (…) that they occupy (…)" mean?

Line 100 ff. *Ocean platforms can be actively driven, they might be buoyantly floating moored designs, and most designs have some sort of active ocean motion control.* 1. To me the term ocean platform seems very uncommon. It might be better to use floating platform or offshore platform instead. Please make sure to be consistent with this throughout the paper. 2. I think that the term driving is not correct here. 3. At least in the context of FOWT, active stabilisation systems are sometimes utilised, but are rare in general to my knowledge.

Line 115 ff. *In what follows, however, we will always line up the platforms/turbines such that only symmetric degrees need to be considered (…)* Please clarify what 'line up the platform' means. An easier way to write this would be. In the following, we focus on surge, sway and yaw motions. Therefore, the system of equation can be reduced to the related DOFs.

Line 117: *suge:* typo

Line 155-163: This text summarises the goal of the paper nicely. It should be located somewhere in the introduction.

Equation 15:Please define the meaning of ^ (over the x).

Line 201: Please clarify what exactly is meant by *turbine simulations.*